# Pulse Oximetry as an Aid to Rule Out Pneumonia among Patients with a Lower Respiratory Tract Infection in Primary Care

**DOI:** 10.3390/antibiotics12030496

**Published:** 2023-03-02

**Authors:** Chloé Fischer, José Knüsli, Loïc Lhopitallier, Estelle Tenisch, Marie-Garance Meuwly, Pauline Douek, Jean-Yves Meuwly, Valérie D’Acremont, Andreas Kronenberg, Isabella Locatelli, Yolanda Mueller, Nicolas Senn, Noémie Boillat-Blanco

**Affiliations:** 1Infectious Diseases Service, Lausanne University Hospital and University of Lausanne, 1011 Lausanne, Switzerland; 2Gare10 Lausanne General Practice, 1011 Lausanne, Switzerland; 3Department of Radiology, Lausanne University Hospital and University of Lausanne, 1011 Lausanne, Switzerland; 4Digital Global Health Department, Centre for Primary Care and Public Health (Unisanté), University of Lausanne, 1011 Lausanne, Switzerland; 5Medix General Practice, 3010 Bern, Switzerland; 6Institute for Infectious Diseases, University Bern, 3010 Bern, Switzerland; 7Department of Education, Research, and Innovation, Centre for Primary Care and Public Health (Unisanté), University of Lausanne, 1011 Lausanne, Switzerland; 8Department of Family Medicine, Centre for Primary Care and Public Health (Unisanté), University of Lausanne, 1011 Lausanne, Switzerland

**Keywords:** pneumonia, infiltrate, chest X-ray, antibiotics, primary care, vital signs, pulse oximetry, lower respiratory tract infections, clinical decision rule

## Abstract

Guidelines recommend chest X-rays (CXRs) to diagnose pneumonia and guide antibiotic treatment. This study aimed to identify clinical predictors of pneumonia that are visible on a chest X-ray (CXR+) which could support ruling out pneumonia and avoiding unnecessary CXRs, including oxygen saturation. A secondary analysis was performed in a clinical trial that included patients with suspected pneumonia in Swiss primary care. CXRs were reviewed by two radiologists. We evaluated the association between clinical signs (heart rate > 100/min, respiratory rate ≥ 24/min, temperature ≥ 37.8 °C, abnormal auscultation, and oxygen saturation < 95%) and CXR+ using multivariate analysis. We also calculated the diagnostic performance of the associated clinical signs combined in a clinical decision rule (CDR), as well as a CDR derived from a large meta-analysis (at least one of the following: heart rate > 100/min, respiratory rate ≥ 24/min, temperature ≥ 37.8 °C, or abnormal auscultation). Out of 469 patients from the initial trial, 107 had a CXR and were included in this study. Of these, 26 (24%) had a CXR+. We found that temperature and oxygen saturation were associated with CXR+. A CDR based on the presence of either temperature ≥ 37.8 °C and/or an oxygen saturation level < 95% had a sensitivity of 69% and a negative likelihood ratio (LR−) of 0.45. The CDR from the meta-analysis had a sensitivity of 92% and an LR− of 0.37. The addition of saturation < 95% to this CDR increased the sensitivity (96%) and decreased the LR− (0.21). In conclusion, this study suggests that pulse oximetry could be added to a simple CDR to decrease the probability of pneumonia to an acceptable level and avoid unnecessary CXRs.

## 1. Introduction

Lower respiratory tract infections (LRTIs) are frequently encountered in general practice and are among the leading causes of antibiotic prescription. Unfortunately, a large proportion (from 50% to 80%) of these prescriptions are inappropriate, driving antibiotic resistance [1,2,3,4,5]. These data highlight the need for additional and easy-to-use diagnostic tools to support clinical decisions in primary care and optimize prescribing practices. However, there is no existing consensus to assess the appropriateness of an antibiotic prescription as it is influenced by many variable factors [3]. Guidelines recommend chest X-rays (CXRs) in patients presenting clinical pneumonia, which is defined as an acute cough and at least one of the following criteria: a history of fever of more than four days or dyspnea or tachypnea or focal abnormal lung auscultation to confirm the diagnosis of pneumonia and guide antibiotic therapy [6,7,8]. However, radiology is frequently performed without a clear indication, and this exposes patients to ionizing radiation [9]. Additionally, the limited agreement in CXR interpretation between general practitioners (GPs) and radiologists is limited, raising questions about the utility of radiology in primary care [10]. A recent systematic review and meta-analysis suggested that adults with acute respiratory infections who have normal vital signs and a normal lung auscultation in the outpatient setting have a very low probability (0.4% to 0.8%) of having community-acquired pneumonia and do not need additional diagnostic testing [11]. If applied to all patients with a suspicion of pneumonia in primary care, such a CDR could significantly decrease the number of unnecessary CXRs performed, thereby limiting exposure to radiation, reducing unnecessary antibiotic prescriptions, saving costs, and minimizing environmental impacts. To the best of our knowledge, although several studies have examined the performance of pulse oximetry to predict pneumonia severity, only a few have evaluated the value of peripheral oxygen saturation levels in diagnosing radiological pneumonia that is visible on a CXR. Additionally, only one study has done so in a primary care setting among patients with coughs attributed to an LRTI. No study evaluated the usefulness of pulse oximetry in primary care among patients with clinical pneumonia, the only subgroup of patients with LRTI for which a CXR and, accordingly, an antibiotic is recommended [9,12]. This study aims to evaluate the value of peripheral oxygen saturation, an easy-to-measure parameter, in addition to other vital signs and clinical examination to rule out pneumonia visible on CXR (CXR+) among patients with a clinical suspicion of pneumonia in primary care.

## 2. Material and Methods

### 2.1. Study Design, Setting, and Population

This is a secondary analysis of a three-group cluster randomized trial which evaluated the impact of using procalcitonin and lung ultrasonography to reduce antibiotic prescriptions for clinical pneumonia in comparison to typical care across 60 primary care practices in Switzerland [13]. General practitioners included adult patients (aged 18 or over) presenting with clinical pneumonia as defined by the European guidelines for community-acquired pneumonia (an acute cough and at least one of the following: focal abnormal finding upon lung auscultation, dyspnea, tachypnea, or a history of fever > 4 days) between September 2018 and March 2020 [6]. The exclusion criteria were: a previous prescription of antibiotics for the current episode, a working diagnosis of acute sinusitis or of a non-infective disorder, a previous episode of chronic obstructive pulmonary disease exacerbation treated with antibiotics during the last 6 months, known pregnancy, severe immunodeficiency (untreated HIV infection with CD4 count < 200 cells/mm^3^, solid organ transplant receiver, and neutropenia (<1000 cells/μL), treatment with corticosteroids (a dose equivalent to 20 mg prednisone/day for >28 days)), a decision by the GP to admit the patient, a GP unavailable for performing the study, and a patient unable to provide informed consent [14]. General practitioners were free to request CXRs. We did not give recommendations on CXRs. In this secondary analysis, only patients who had a CXR performed at the time of inclusion and whose vital signs had all been measured were included.

During the enrollment process, GPs collected data on patients’ demographics, comorbidities, clinical symptoms, and vital signs, including oxygen saturation, in an electronic case report form using REDCap^®^ (Research Electronic Data Capture) [15]. Temperature was obtained by tympanic measurement.

Written informed consent was obtained from all participants. The study was approved by the Swiss Ethics Committee of the canton of Vaud and Bern (2017-01246).

### 2.2. X-ray Interpretation

After the CXRs were performed, GPs recorded their interpretation of the images by filling out a standardized report form that assessed the presence or the absence of an abnormal finding in keeping with pneumonia [6,16]. At the end of the study, digital CXRs were collected and reviewed by two experienced radiology specialists from the University Hospital of Lausanne (ET and JYM) who were blinded to the patient’s clinical presentation and who completed the same standardized report form as the GPs. A third radiologist (P.D.) solved discrepancies. 

### 2.3. Statistical Analysis

Differences in proportion between patients with pneumonia visible on CXR (CXR+) and those without (CXR−) were evaluated by Pearson’s chi-squared test or Fisher’s exact test where appropriate. To account for the generalizability of our findings outside the sample of interest, differences between patients who had a CXR and those who did not (in the initial clinical trial) were also evaluated. The association of vital signs (including peripheral capillary oxygen saturation) and lung auscultation with a CXR+ was evaluated using univariate logistic regression. Vital signs were dichotomized at predefined cut-off points regarded as the boundary between normal and abnormal [9]. We chose cut-offs which were frequently used in previous studies conducted in the same setting [11]. We defined abnormal lung auscultation as any focal abnormal finding upon lung auscultation. A multivariate logistic regression model was subsequently determined using backward selection to minimize the Akaike information criterion, which was performed using the *MASS* package for R [17]. Diagnostic performance measures including sensitivity, specificity, overall diagnostic accuracy (the correctly classified proportion), positive and negative predictive values [PPV, NPV], positive and negative likelihood ratios [LR+, LR−], and the area under the receiver operating characteristic curve (AUROC) were calculated for identified clinical signs. We then derived a CDR using these signs to exclude the CXR+ and evaluated its performance on our study sample. We also tested a clinical decision rule derived from a large meta-analysis by Marchello et al., defined as the presence of at least one of the following: heart rate > 100/min, respiratory rate ≥ 24/min, temperature ≥ 37.8 °C, or an anomaly found upon lung auscultation [11]. Additionally, we estimated the added value of pulse oximetry to the clinical rule by Marchello et al. The comparisons for differences in sensitivity were made using the Wald interval with a Bonett–Laplace adjustment, while Wald intervals were used to compare ratios of LR− between two clinical decision rules, as per the formulae given by Roldán-Nofuentes [18,19]. 

The kappa coefficient was calculated to measure the inter-rater agreement in the CXR interpretation between the radiologists’ consensus and the GPs and between the two radiologists. The proportional agreement in positive cases (presence of pneumonia) and in negative cases (absence of pneumonia), as well as the concordance (in both pneumonia and non-pneumonia patients), was calculated between the radiologists’ consensus and the GPs and between the two radiologists. All statistical analyses were performed using R Statistical Software version 4.1.1 [20]. A *p*-value < 0.05 was considered statistically significant. For the sensitivity differences (or ratios of negative likelihood ratios), a 95% confidence interval not containing 0 (or 1, respectively) was considered statistically significant.

## 3. Results

### 3.1. Study Population

Out of a total of 469 patients included in the initial trial, 130 (28%) had a CXR, of which 23 patients were not included in the analysis due to missing vital signs measurements. Among the 107 remaining patients who were included in this study, a majority were female (61%), and approximately one-third (32%) were older than 65 and had at least one comorbidity (27%). Overall, 26/107 (24%) patients had a CXR+. Patients CXR+ were less often female (42% versus 67%, *p* = 0.047), more often older than 65 years (54% versus 25%, *p* = 0.011), and had less asthma (0% versus 24%, *p* = 0.006) when compared to patients CXR−. Additionally, a greater proportion of CXR+ patients had a history of fever (89% versus 63%, *p* = 0.027), a temperature ≥ 37.8 °C (46% versus 14%, *p* = 0.001), and an oxygen saturation < 95% (46% versus 21%, *p* = 0.024) when compared to CXR− patients (Table 1).

### 3.2. Generalizability of the Findings to the Population of the Original Cluster Randomized Trial

The patients included in this secondary analysis were found to have similar demographics and comorbidities compared to those who were not included. However, a larger proportion of patients had a respiratory rate ≥ 24/min (23% vs. 15%, *p* = 0.043) and an abnormal lung auscultation (70% vs. 41%, *p* < 0.001) compared to those who were not included. This information is presented in Appendix A.

### 3.3. Association of Clinical Signs with Pneumonia Visible on Chest X-rays (CXR+)

Of the five tested clinical signs (lung auscultation and vital signs apart from blood pressure as no patients with CXR+ had low blood pressure), a temperature ≥ 37.8 °C (odds ratio [OR] 5.5, 95% confidence interval [95% CI] 2 to 15, *p* = 0.001), and an oxygen saturation < 95% (3.2, 1.3 to 8.3, *p* = 0.014) were found to be associated with CXR+ in the univariate logistic regression (Table 2).

Using the same covariates, the backward selection algorithm determined a multivariate model that included a temperature ≥ 37.8 °C (OR 5.0, 95% CI 1.8 to 14, *p* = 0.002) and an oxygen saturation < 95% (2.9, 1.0 to 7.9, *p* = 0.037) to predict CXR+.

### 3.4. Diagnostic Performance

Given our aim of using clinical signs to exclude CXR+, we created a clinical decision rule from the two identified predictors of pneumonia: the presence of either a temperature ≥ 37.8 °C or an oxygen saturation < 95%. This clinical decision rule had a sensitivity of 69% (95% CI 48% to 86%), a NPV of 88% (95% CI 77% to 94%), and an LR− of 0.45 (95% CI 0.25 to 0.81) with an overall diagnostic accuracy of 69% (95% CI 59% to 78%) (Table 3). By using this rule, CXRs could be avoided in 60% (64/107) of patients at the cost of missing 31% (8/26) of pneumonia cases. The calculated AUROC for this clinical decision rule was 0.69 (95% CI 0.59 to 0.80).

For comparison, we evaluated the diagnostic performance of the clinical rule by Marchello et al. in our population [11]. These criteria had a sensitivity of 92% (95% CI 75% to 99%), a NPV of 89% (95% CI 67% to 99%), and an LR− of 0.37 (95% CI 0.09 to 1.48). However, they had a low specificity of 21% (95% CI 13% to 31%). The use of Marchello’s criteria could have avoided CXRs in 18% (19/107) of patients at the cost of missing 7.7% (2/26) of pneumonia cases. Compared to our clinical decision rule, the improvement in sensitivity was significant (sensitivity difference of 23%, 95% CI 1% to 42%) but the improvement in LR− was not (LR−ratio of 0.82, 95% CI 0 to 2.02). The calculated AUROC for the clinical rule by Marchello et al. was 0.57 (95% CI 0.50 to 0.64).

Since our regression analysis showed that the peripheral oxygen saturation was a predictor of CXR+, we added oxygen saturation to the clinical rule by Marchello et al. This increased the sensitivity and decreased the LR− (sensitivity 96% [95% CI 80% to 100%], specificity 19% [11% to 29%], NPV 94% [70% to 100%], and LR−0.21 [0.03 to 1.50]) compared to the original validated rule. However, these improvements were not statistically significant (sensitivity difference 4%, 95% CI−0.08 to 0.16 and LR−ratio 0.57, 95% CI−0.22 to 1.36). 

### 3.5. Agreement in Chest X-ray Interpretation 

The inter-rater agreement between the radiologists’ consensus and the GP’s interpretation of the CXRs was kappa = 0.62 (95% CI 0.46 to 0.78). The two radiologists who read the X-rays had an agreement (on the presence or the absence of an abnormal finding consistent with pneumonia) of kappa = 0.52 (95% CI 0.33 to 0.72) between them [6,16]. We found a concordance (in both pneumonia and non-pneumonia patients) of 84% (95% CI 76 to 90%) between the radiologists’ consensus and the GP’s interpretation and of 85% (95% CI 77 to 91%) between the two radiologists. The proportional agreement in positive cases (presence of pneumonia) was 73% and in 89% in negative cases (absence of pneumonia) between the radiologists’ consensus and the GP’s interpretation, and 60% and 91% between the two radiologists, respectively.

## 4. Discussion

### 4.1. Main Results

Among patients with a clinical suspicion of pneumonia in primary care and with a decision by the GP to perform a CXR, a temperature ≥ 37.8 °C and an oxygen saturation < 95% were both found to be predictors of pneumonia that was visible on a CXR. A clinical decision rule that included either a temperature ≥ 37.8 °C or an oxygen saturation < 95% had limited sensitivity and a moderate LR−, allowing for the avoidance of a CXR in almost two-thirds of patients at the cost of missing a third of the pneumonia visible on a CXR. In comparison, the clinical rule by Marchello et al., which is based on three vital signs and lung auscultation, had better sensitivity and a similar LR−, but its low specificity allowed for the avoidance of CXRs only in a minority of patients. However, its high sensitivity allowed it to miss four times fewer pneumonia cases than our “fever or hypoxemia” clinical decision rule. Adding oxygen saturation as an additional criterion to the clinical rule by Marchello et al. further lowered the LR− to a helpful level.

Due to the selection bias in our population, the prevalence of CXR+ in our study was 24%, which is five times higher than what is usually reported in primary care (between 4% and 6%) [21,22,23,24]. Applying our clinical rule with an LR− of 0.45, the post-test probability of pneumonia was reduced by half (from 24% to 12%) in the absence of fever and hypoxemia. Applying the clinical rule by Marchello et al., complemented by pulse oximetry with a LR− of 0.21, and in the absence of fever, tachycardia, tachypnea, hypoxemia, and abnormal lung auscultation, the post-test probability of pneumonia decreased by four times (from 24% to 6%). 

Deciding on probability thresholds to recommend testing to rule out pneumonia is challenging. The GRACE consortium (Network of Excellence focusing on community-acquired lower respiratory tract infections) decided on cut-off levels of three risk groups based on their clinical judgment and a review of acceptability of false negative results in other diagnostic studies in primary care: a low-risk group (probability < 2.5%), an intermediate-risk group (2.5–20%), and high-risk group for pneumonia (>20%) [24]. A quantitative study evaluated the test decision thresholds for the management of patients with an acute cough in primary care among 256 physicians and showed a higher threshold compared with the GRACE consortium, with a cut-off of 10% considered a low-risk group of pneumonia. Below this cut-off, physicians would not recommend diagnostic tests nor initiate antibiotics [22]. These data suggest that the clinical rule by Marchello et al. complemented by pulse oximetry might decrease the probability of pneumonia below the test threshold of clinicians.

The agreement in the interpretation of the CXRs between the two radiologists was moderate, likely due to poor image quality and the fact that radiologists were blinded to all clinical information, but was otherwise in line with previous studies, showing a kappa value of 0.45 [24]–0.53 [10]. The agreement in the interpretation of the CXRs between GPs and the two radiologists was good in our study, and it was also in accordance with a previous study that demonstrated a kappa value of 0.77 [25]. As in previous studies, we also observed a lower proportional agreement in positive cases (presence of pneumonia visible on CXR) than in negative cases (absence of pneumonia on CXR) between the two radiologists and between the GPs and the radiologist [25,26].

### 4.2. Comparison with Other Studies

Multiple other studies discussed the use of pulse oximetry in primary care or in the emergency department to evaluate the severity of pneumonia [27,28,29]. However, to the best of our knowledge, only a few studies evaluated the value of oxygen saturation as an aid for the diagnosis of pneumonia in general practice [9]. In a large cohort (N = 28′883), patients with an acute cough attributed to an LRTI were recruited from 5′222 UK practices. Only a minority of patients (2.5%) had a CXR, 16% of which had pneumonia visible on a CXR. In line with our results, this study showed that a temperature > 37.8 °C (relative risk 2.6; 95% CI 1.5–4.8) and an oxygen saturation < 95% (relative risk 1.7; 95% CI 1.0–3.1) were both predictors of pneumonia visible on a CXR. It also identified two additional clinical signs associated with pneumonia visible on a CXR: a heart rate > 100/min (relative risk 1.9; 95% CI 1.1–3.2) and crackles on auscultation (relative risk 1.8; 95% CI 1.1–3.0). Compared to the population of our initial clinical trial in Switzerland, the patients included in this secondary analysis had abnormal auscultation more often (70% vs. 41%), representing a selection bias that may have weakened the association of this clinical sign with pneumonia visible on a CXR. In the aforementioned large UK cohort, the NPV, LR−, and AUROC of the presence of one of the following criteria to diagnose pneumonia visible on CXR: temperature > 37.8 °C, abnormal auscultation (crackles), oxygen saturation < 95% and heart rate > 100/min were similar to our clinical rule based on only two clinical criteria (temperature > 37.8 °C and oxygen saturation < 95%) (93% versus 88%, 0.42 versus 0.45 and 0.68 versus 0.69, respectively). However, the sensitivity of the UK-based clinical rule was higher compared with our two clinical criteria decision rule (86% vs. 69% in our cohort) but was similar to the clinical rule by Marchello et al. (86% vs. 92% in the rule by Marchello et al.). The lower sensitivity of our “two vital signs” clinical rule may be explained by the inclusion of a population with more severe LRTIs compared to the large UK study. Indeed, we included patients with clinical pneumonia and not only cough, and the proportion of patients with pneumonia visible on CXR was higher. A retrospective case-control study including patients in the emergency department of a US hospital derived a three-step algorithm (elevated temperature, tachycardia, and hypoxemia) using a classification tree analysis. It had a sensitivity similar to our two clinical signs rule (71% vs. 69%), but a higher specificity (79% vs. 69%). Another retrospective study conducted in a US emergency department among patients with respiratory symptoms showed that a clinical decision rule based on the presence of any abnormality of the vital signs (temperature > 38 °C, heart rate > 100/min, respiration rate > 20/min or pulse oximetry < 95%) had a sensitivity at 90% and an LR− at 0.13: better than our two clinical signs rule but similar to the clinical rule by Marchello et al. complemented by pulse oximetry [30]. A small, retrospective case-control study conducted in a veterans’ nursing home evaluated pulse oximetry in pneumonia and showed a sensitivity of 80% and a specificity of 91% for an oxygen saturation < 94% [22,31].

Several studies analyzed clinical prediction rules to rule pneumonia in or out in primary care. An individual patient data meta-analysis evaluated six prediction models for pneumonia visible on a CXR in primary care based on signs and symptoms [32]. The model by van Vugt et al., which is based on six predictors, had the highest discriminative accuracy with an AUROC of 0.79 [24]. In comparison, our clinical rule based on two vital signs had a lower discriminative accuracy with an AUROC of 0.69, suggesting that more criteria might be safer. According to the authors, the “signs and symptoms” model of van Vugt is the best candidate for primary care use. However, it is based on subjective symptoms and a detailed lung auscultation, which is often non-reproducible between clinicians [33]. We could not test this “signs and symptoms” rule in our study sample as we did not collect information on coryza nor enough details on abnormalities in lung auscultation. Another recent meta-analysis evaluated studies using signs and symptoms to rule out pneumonia in primary care and identified the clinical rule by Marchello et al.: the presence of abnormal vital signs (heart rate > 100/min or respiratory rate ≥ 24/min or temperature ≥ 37.8 °C) or abnormal lung auscultation had an LR− of 0.10 (95% CI 0.07 to 0.13) and an AUROC of 0.92 [11]. As detailed in the results section, we tested this clinical rule in our patient sample and found a higher LR− (0.37) and a lower AUROC (0.57). This discrepancy might be due to our study population, in whom the prevalence of pneumonia is higher than in the meta-analysis by Marchello et al. (24% vs. 4%). 

What is the added value of our study? Our study extends the knowledge on the usefulness of oxygen saturation to patients with clinical pneumonia in primary care for use in a simple clinical decision rule based on two vital signs, i.e., temperature and oxygen saturation, that could potentially avoid numerous CXRs, reducing patient irradiation and saving costs. Our study also be integrated into the clinical rule by Marchello et al., preventing fewer CXRs but ensuring the safety of patients by missing only few instances of pneumonia visible on a CXR. 

### 4.3. Strengths and Limitations 

The main strength of our study lies in the data collection on demographics, comorbidities, and vital signs, which were recorded in almost all patients. Only a minority had missing data for vital signs. Another strength lies in the inclusion of patients with clinical pneumonia, the only LRTI for which a CXR and, accordingly, an antibiotic is recommended [6]. Our outcome of interest, the diagnosis of pneumonia visible on CXR, was rigorously established using the radiological reference standard with the help of three experienced radiologists. Of note, our results also show the difficulties in CXR interpretation, with a moderate agreement in interpretation between the two radiologists. Furthermore, our study population had similar characteristics (61% female, 32% older than 65, and 27% with any comorbidity) to large previous cohorts of patients with LRTIs in primary care, highlighting its representativeness [9,24].

The main limitation of our study is the selection bias of our study population since the CXRs were performed in selected patients based on each GP’s clinical judgment, rather than on reproducible criteria. The included patients had more tachypnea and abnormal lung auscultations compared to those not included, which suggests a population with a higher prevalence of pneumonia; the high proportion of patients with pneumonia visible on CXR in our sample confirms this bias. This may affect the generalization of our results. However, our study population was otherwise similar to the whole population of the initial clinical trial in terms of demographics and comorbidities. Another limitation is the small study sample size of 107 patients. A larger sample size could have identified additional clinical signs or symptoms that predict a diagnosis of pneumonia and could have provided more confidence in the results by reducing the size of the 95% confidence intervals. Another limitation of our study was the imperfect gold standard for the definition of CAP. Indeed, we defined CAP as the presence of an infiltrate on a CXR among patients with LRTIs. The presence of an infiltrate does not provide information on its temporality, and it may be the sequella of another lung pathology. In addition, a chest x-ray is an imperfect tool for identifying a lung infiltrate. We do not have information on the ethnicity of the patients. As darker-skinned patients are more likely to have falsely elevated oxygen saturation, our data may not be easily extrapolated to darker-skinned patients [34].

### 4.4. Implications for Practice and Future Research

Compared with other “signs and symptoms” clinical decision rules recommended for use in primary care such as the van Vugt and Marchello criteria, our clinical rule, based on only two vital signs, i.e., temperature and oxygen saturation, might be an easy-to-perform alternative to rule out pneumonia in primary care. It is reproducible and less prone to variation between clinicians than the respiratory symptoms and lung auscultation which are among the van Vugt and Marchello criteria. However, it decreases the post-test probability of pneumonia visible on a CXR to a higher level (12%) than the test threshold (10%) of clinicians according to a quantitative study among 256 physicians, which jeopardizes its acceptability [22]. Adding pulse oximetry to the clinical rule by Marchello et al. decreases the probability of pneumonia below the test threshold of clinicians and might increase its acceptability by physicians.

In conclusion, pulse oximetry is an interesting and affordable additional tool for ruling out pneumonia visible on a CXR which can be added to easy-to-measure vital signs in primary care. Pulse oxymetry increases the sensitivity and therefore the safety of available clinical rules based on vital signs alone and could help convince physicians that additional tests or antibiotics are not necessary. This has the potential to avoid unnecessary irradiation, save costs, and decrease antibiotic prescriptions. However, the impact of pulse oximetry in a clinical decision rule on the safety of patients and antibiotic prescriptions should be tested in a clinical trial.

## Figures and Tables

**Table 1 antibiotics-12-00496-t001:** Patients’ characteristics according to the presence of pneumonia visible on chest X-ray.

	All	No pneumonia on Chest X-ray CXR−	Pneumonia on Chest X-ray CXR+	*p*
	N = 107	N = 81 (76%)	N = 26 (24%)	
**Demographics and comorbidities**				
Female	65 (61)	54 (67)	11 (42)	0.047
Age ≥ 65 years	34 (32)	20 (25)	14 (54)	0.011
Active smoker	24 (22)	19 (24)	5 (19)	0.790
Any comorbidity	28 (27)	24 (30)	4 (15)	0.213
Chronic obstructive pulmonary disease	8 (7.5)	5 (6.2)	3 (12)	0.399
Asthma	19 (18)	19 (24)	0 (0.0)	0.006
Other comorbidity *	5 (4.8)	3 (3.8)	2 (7.7)	0.597
**Symptoms and signs**				
Sputum production	74 (70)	60 (74)	14 (56)	0.141
History of fever	74 (69)	51 (63)	23 (89)	0.027
History of dyspnoea	73 (69)	60 (75)	13 (50)	0.032
History of chest pain	46 (43)	38 (47)	8 (31)	0.176
Heart rate > 100/min	17 (16)	11 (14)	6 (23)	0.354
Temperature ≥ 37.8 °C	23 (22)	11 (14)	12 (46)	0.001
Oxygen saturation < 95%	29 (27)	17 (21)	12 (46)	0.024
Respiratory rate ≥ 24/min	25 (23)	19 (24)	6 (23)	1.000
Hypotension **	8 (7.5)	8 (10)	0 (0.0)	0.195
CRB-65 ≥ 1 point	39 (36)	25 (31)	14 (54)	0.060
Abnormal lung auscultation	75 (70)	55 (68)	20 (77)	0.530

Values are *n* (%). * Other comorbidity: heart failure (0/107, 0.0%), diabetes (3/107, 2.9%), active malignancy (2/107, 1.9%), chronic kidney disease (0/107, 0.0%), or human immunodeficiency virus infection (0/107, 0.0%). ** Hypotension: systolic blood pressure ≤ 90 mmHg or diastolic blood pressure ≤ 60 mmHg.

**Table 2 antibiotics-12-00496-t002:** Logistic regression analyses: association of vital signs and lung auscultation with pneumonia visible on chest X-ray (CXR+).

	Univariate Analysis		Multivariate Analysis	
	OR [95% CI]	*p*	OR [95% CI]	*p*
Abnormal lung auscultation	1.6 [0.59, 4.7]	0.384		
Temperature ≥ 37.8 °C	5.5 [2.0, 15]	0.001	5.0 [1.8, 14]	0.002
Oxygen saturation < 95%	3.2 [1.3, 8.3]	0.014	2.9 [1.1, 7.9]	0.037
Heart rate > 100/min	1.9 [0.60, 5.7]	0.254		
Respiratory rate ≥ 24/min	0.98 [0.32, 2.7]	0.968		

OR: odds ratio. 95% CI: 95% confidence interval. Low blood pressure was not analyzed as no patient had low blood pressure among patients with radiological pneumonia.

**Table 3 antibiotics-12-00496-t003:** Diagnostic performance of selected vital signs according to the results of the multivariate logistic regression and by the clinical rule by Marchello et al. for pneumonia visible on chest X-rays (CXR+).

	*n* (%) N = 107	Sensitivity	Specificity	Diagnostic Accuracy	NPV	PPV	LR−	LR+	Prevented X-rays N = 107	Missed Pneumonia Visible on CXR N = 26
Temperature ≥ 37.8 °C AND oxygen saturation < 95%	9 (8%)	23% [0.09, 0.44]	96% [0.90, 0.99]	79% [0.70, 0.86]	80% [0.70, 0.87]	67% [0.30, 0.93]	0.80 [0.64, 0.99]	6.23 [1.68, 23.18]	98 (92%)	20 (77%)
Oxygen saturation < 95%	29 (27%)	46% [0.27, 0.67]	79% [0.69, 0.87]	71% [0.61, 0.79]	82% [0.72, 0.90]	41% [0.24, 0.61]	0.68 [0.47, 0.99]	2.20 [1.22, 3.98]	78 (73%)	14 (54%)
Temperature ≥ 37.8 °C	23 (21%)	46% [0.27, 0.67]	86% [0.77, 0.93]	77% [0.67, 0.84]	83% [0.74, 0.91]	52% [0.31, 0.73]	0.62 [0.43, 0.90]	3.40 [1.71, 6.77]	84 (79%)	14 (54%)
Temperature ≥ 37.8 °C OR oxygen saturation < 95%	43 (40%)	69% [0.48, 0.86]	69% [0.58, 0.79]	69% [0.59, 0.78]	88% [0.77, 0.94]	42% [0.27, 0.58]	0.45 [0.25, 0.81]	2.24 [1.48, 3.40]	64 (60%)	8 (31%)
Clinical rule by Marchello et al. *	88 (82%)	92% [0.75, 0.99]	21% [0.13, 0.31]	38% [0.29, 0.48]	89% [0.67, 0.99]	27% [0.18, 0.38]	0.37 [0.09, 1.48]	1.17 [1.00, 1.37]	19 (18%)	2 (7.7%)
Clinical rule by Marchello et al. * OR oxygen saturation < 95%	91 (85%)	96% [0.80, 1.00]	19% [0.11, 0.29]	37% [0.28, 0.47]	94% [0.70, 1.00]	27% [0.19, 0.38]	0.21 [0.03, 1.50]	1.18 [1.04, 1.34]	16 (15%)	1 (3.8%)

Values are *n* (%), percentage [95% confidence interval] or value [95% confidence interval]. NPV: negative predictive value; PPV: positive predictive value; LR−: negative likelihood ratio; LR+: positive likelihood ratio. * Clinical rule by Marchello et al.: presence of at least one of the following criteria: heart rate > 100/min; respiratory rate ≥ 24/min; temperature ≥ 37.8 °C; or anomaly found upon lung auscultation.

## Data Availability

All relevant data are available via the Zenodo repository (10.5281/zenodo.4032527).

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
