# Peer review of "Pulse Oximetry as an Aid to Rule Out Pneumonia among Patients with a Lower Respiratory Tract Infection in Primary Care"

_antibiotics, 2023, doi:10.3390/antibiotics12030496_

Round 1

Reviewer 1 Report

I read, Pulse oximetry as an aid to rule out pneumonia among patients with a lower respiratory tract infection in primary care, with interest. In this manuscript, the authors aimed to identify clinical predictors of pneumonia visible on chest Xray (CXR+), including oxygen saturation, which could support ruling out pneumonia and avoid CXRs.

 I have some questions and suggestion.

1) Can you explain why this study is new or telling new things?

2) Discussion is rather weak. Please compare the results of this study with the results of other studies for a more in-depth discussion.

3) Why did the body temperature criterion in your study use ≥ 37.8°C or an oxygen saturation < 95%? Why weren't other cut-off values used?

4) Please provide more data on the importance of physicians and healthcare professionals around the world to recognize of new strategy for rule out pneumonia among patients with a lower respiratory tract infection in primary care.

5) Please add more limitation in your study.

Minor

Line 118-128, Table 2: The font size is not the same. The author should adjust the font size to be the same for all manuscript.

Reviewer 2 Report

The article entitled as "Pulse oximetry as an aid to rule out pneumonia among patients with a lower respiratory tract infection in primary care" by Chloé Fischer et al is a very important hot topic that will help the clinicians while  diagnosing and treating the patients suffering from pneumonia and other LRTIs.My suggestion is to add the future perspective in short words and recheck and remove the typo and grammatical mistakes from the manuscript to further improve the MS.

Reviewer 3 Report

To the authors, 

Fischer et al. conducted a study titled “Pulse oximetry as an aid to rule out pneumonia among patients with a lower respiratory tract infection in primary care”. The authors present an interesting manuscript about the predictive potential of oxygen duration for ruling out pneumonia. While the approach seems appropriate, there are some suggestions: 

1.     For the ICs and ECs, you referred to other manuscript. I think the list of these criteria is not too long and is very important for the context. I would advise to include this in the manuscript. 

2.     It is known that darker skin may interfere with the accuracy of pulse oximeters. Is it possible to provide the ethnical background of your study population as well as information about skin color. Moreover, is it possible to perform corresponding subgroup analyses?

3.     You stated that the included and non-included groups were quite similar. How is it explained that one group was recommended to perform a chest x-ray and the other one not?

4.     “Presumably due to the selection bias in our population, the prevalence of CXR+ in our study is of 24%, which is five time higher than usually reported in primary care (between 4% and 6%).” … This also indicates that the sample is not very representative. 

5.     “Using this 9 rule, we could avoid doing a CXR in 64/107 (60%) patients at the cost of missing 8/26 (31%) 10 pneumonia CXR+. The calculated AUROC for this clinical decision rule was 0.69 (95% CI 11 0.59 to 0.80).” I think that these numbers are actually not very reassuring. Missing 1/3 of pneumonia patients suggests a questionable performance…

6.     It would be interesting analyse subgroups in which a reduced oxygen saturation can be expected (i.e. smokers and COPD patients). How is the performance in these groups. 
